# Environmental permittivity-asymmetric BIC metasurfaces with electrical reconfigurability

Haiyang Hu [1,4], Wenzheng Lu [1,4], Alexander Antonov[1], Rodrigo Berté [1], Stefan A. Maier [2,3] & Andreas Tittl [1] ✉

Achieving precise spectral and temporal light manipulation at the nanoscale remains a critical challenge in nanophotonics. While photonic bound states in the continuum (BICs) have emerged as a powerful means of controlling light, their reliance on geometrical symmetry breaking for obtaining tailored resonances makes them highly susceptible to fabrication imperfections, and their generally fixed asymmetry factor fundamentally limits applications in reconfigurable metasurfaces. Here, we introduce the concept of environmental symmetry breaking by embedding identical resonators into a surrounding medium with carefully placed regions of contrasting refractive indexes, activating permittivity-driven *quasi*-BIC resonances ($\varepsilon$-$q$BICs) without altering the underlying resonator geometry and unlocking an additional degree of freedom for light manipulation through active tuning of the surrounding dielectric environment. We demonstrate this concept by integrating polyaniline (PANI), an electro-optically active polymer, to achieve electrically reconfigurable $\varepsilon$-$q$BICs. This integration not only demonstrates rapid switching speeds and exceptional durability but also boosts the system's optical response to environmental perturbations. Our strategy significantly expands the capabilities of resonant light manipulation through permittivity modulation, opening avenues for on-chip optical devices, advanced sensing, and beyond.

In modern optical physics, permittivity is one of the principal determinants of light-matter interactions, and constitutes a crucial degree of freedom for designing and engineering optical systems and components[1–3]. The artificial engineering of permittivity at interfaces has led to the development of metasurfaces, revolutionizing nanophotonics by enabling subwavelength light manipulation[4–6]. Among versatile metasurface concepts, bound states in the continuum (BIC) metasurfaces stand out as a powerful platform for realizing high-quality resonance modes[7,8], demonstrating the transformative potential of permittivity engineering in driving progress in the fields of nonlinear optics[9,10], sensing[11,12], and lasing[13].

BICs have attracted growing attention in nanophotonics due to their strong photon localization at the nanoscale[7,8]. BICs are distinguished into two primary categories, each defined by the mechanism through which the eigenmodes of the structures evade coupling with the radiative continuum. The first category, known as accidental BICs, arises under the Fridrich-Wintgen scenario[14], where the coupling with radiative waves is suppressed through the tuning

[1]Chair in Hybrid Nanosystems, Nanoinstitute Munich, Faculty of Physics, Ludwig-Maximilians-Universität München, Königinstraße 10, München, Germany. [2]School of Physics and Astronomy, Monash University Clayton Campus, Melbourne, Victoria, Australia. [3]The Blackett Laboratory, Department of Physics, Imperial College London, London, United Kingdom. [4]These authors contributed equally: Haiyang Hu, Wenzheng Lu. ✉e-mail: Andreas.Tittl@physik.uni-muenchen.de

of system parameters. The second category, symmetry-protected BICs, results from the preservation of spatial symmetries, such as reflection or rotation, which inherently restrict the coupling between the bound state and the continuum due to symmetry incompatibility[15,16]. True symmetry-protected BICs are theoretical entities of infinitely high quality ($Q$) factors and vanishing resonance width, which can turn into a *quasi*-BIC ($q$BIC) through the introduction of a finite coupling to the radiation continuum, resulting in a finite $Q$ factor and an observable resonance in far-field spectra[17]. Such coupling can be realized through minor geometric perturbations in the symmetry of resonators within the unit cell, for instance, via changes in the length, height, relative angle, or area of the constituent resonators, etc[9,12,15,18].

The generation of $q$BICs in geometrically modulated systems strongly relies on the geometrical design, and precision of current lithographic techniques[19,20]. Moreover, the fixed geometry of the resonators after fabrication limits the possibilities of harnessing dynamic $q$BICs for potential applications in optical modulation[21,22], dynamic sensors[23], and light guiding[24,25].

An alternative solution involves leveraging the permittivity of individual unit cell components to induce $q$BIC resonances, a strategy that circumvents the need for precise modifications of geometric asymmetry[26–29]. In this case, the asymmetry parameter is given by the difference in the intrinsic permittivity within the unit cell, allowing for the induction of $q$BIC resonances through perturbations in the permittivity symmetry of resonators.

Compared to the intrinsic permittivity of construction materials used for the nanostructures, the permittivity of environmental media can be more conveniently engineered. In this work, we introduce a design concept that leverages environmental permittivity asymmetry to manipulate $q$BIC resonances. We embed identical resonators into surrounding media with different refractive indexes (RI), effectively introducing an environmental permittivity symmetry breaking to this system. As a practical demonstration, we fabricate a metasurface consisting of two identical dielectric nano-rods per unit cell, where one rod is embedded in PMMA with higher RI, contrasting with its counterpart in air. The different RI surroundings break the symmetry and activate permittivity-asymmetric $q$BICs (ε-$q$BICs), which can respond actively to further changes in RI asymmetry by altering the environment of the uncovered nanorod. The concept of the environmental ε-$q$BICs also opens the possibility of engineering dynamic BICs for active narrowband applications, such as tunable modulators, sensors, filters, and lasers[21]. In particular, we incorporate polyaniline (PANI), an electrically active conductive polymer, which has been widely applied in active metasurfaces[30–33] due to its large RI variation and fast switching speed[34–37], for the realization of electrically reconfigurable BICs. This integration, achieved through in-situ polymer growth on the metasurface, ensures mechanical and electrical durability, facilitating rapid and reliable switching of ε-$q$BIC states[31]. Specifically, we successfully engineer the radiative coupling of the ε-$q$BICs by leveraging the electro-optical response of PANI, where the $q$BIC resonance in the transmittance spectra can be switched between the "ON" and "OFF" states with a fast switching speed of 18.8 ms within low operation voltage range from −0.2 V to +0.6 V. Moreover, we observe a superior cycling stability of over 1000 switching cycles without noticeable degradation. Our reconfigurable ε-$q$BIC metasurface platform unlocks a new degree of freedom in manipulating radiation coupling in BIC-driven systems, which is unachievable with static geometry symmetry-breaking approaches. This active response to the environmental perturbation exploits the synergy derived from integrating optics and electrolyte fluidics on a single chip, especially in the important visible spectral range, which can be transformative in the development of on-demand flat optics and sensing technologies[21,38–40].

## Results and discussion
### Coupling of bound states to the radiation continuum mediated by environmental permittivity

For symmetry-protected BIC metasurfaces, a system of two identical rods in an isotropic environment represents an unperturbed, non-radiative bound state (Fig. 1a). Breaking the in-plane symmetry of the unit cell can induce coupling of the bound state to the radiation continuum, resulting in a $q$BIC with a finite $Q$ factor (Fig. 1b). Conventionally, this coupling channel is opened through breaking the symmetry of the unit cell geometry, for instance by shortening one of the rods, thus removing one of the original mirror plane symmetries and giving rise to geometry-asymmetric *quasi*-BICs (g-$q$BICs). The asymmetric factor here is proportional to the volume change of one of the resonators (Fig. 1a).

Alternatively, changes in environmental permittivity can disrupt the system's symmetry, thereby enabling the coupling of true BIC modes to the radiative continuum (Fig. 1a). In this configuration, two identical nano-rods (within the unit cell) are embedded into heterogeneous surrounding media, each with a different refractive index (RI). Specifically, one nano-rod is placed in a medium with an RI of 1.5 (PMMA), while the other is in the air with an RI of 1.0 (Fig. 1a). This RI contrast disrupts their original destructive interference, transitioning the system from a true BIC to an ε-$q$BIC. This change makes the resonance observable in the far field.

Here, each meta-atom acts as a Mie-resonant nanoparticle that supports various multipole modes, including electric and magnetic dipoles, as well as higher-order multipole modes[16,41]. The scattering efficiency of each meta-atom is influenced by the characteristics of its surrounding medium[42,43]. Through applying multipole analysis, we can gain more insight into the origin of $q$BIC in this design (Supplementary Note 1). The coupling coefficient of the eigenstate with the normally incident light along the $z$-direction with a wave vector $k = \omega/c$ and polarization along **e** is proportional to an overlap integral[15]:

$$m_{\mathbf{e}} \propto \int_{V_1, V_2} \mathbf{J}(\mathbf{r}) \cdot \mathbf{e} e^{i\mathbf{k}\cdot\mathbf{r}} dV, \tag{1}$$

where $\mathbf{J}(\mathbf{r})$ is a displacement current density, $V_1$ and $V_2$ are the volumes of the dielectric nanorods. In the case of symmetry-protected BIC the resonance of the system is described by a pair of antiparallel dipole moments $\mathbf{p}_1 = -\mathbf{p}_2$, which obviously eliminates the coupling: $m_{\mathbf{e}} \propto \mathbf{p}_1 \cdot \mathbf{e} + \mathbf{p}_2 \cdot \mathbf{e} = 0$. Then by introducing PMMA cladding, which breaks one of the mirror plane symmetries along $y$ direction, and performing multipole expansion of the coupling coefficient, one can easily obtain (see details in Supplementary Note 1):

$$m_y \propto P_y - \frac{1}{c}M_x - \frac{i\omega}{6c}Q_{yz}, \tag{2}$$

where the components of electric dipole, magnetic dipole, and electric quadrupole moments are introduced. Consequently, we employ numerical simulation by COMSOL Multiphysics to compare the contributions of all terms to the coupling coefficient for the PMMA-covered model. We integrate the calculated fields separately over both nanorods and PMMA according to Supplementary Equations (3)–(5). As shown in Supplementary Fig. 1c, the main contribution comes from electric dipole moments $P_y$. Nevertheless, the role of the magnetic dipole $M_x$ and electric quadrupole $Q_{yz}$ moments cannot be neglected: their corresponding absolute value at quasi-BIC resonance is only 3 times less than the value of $P_y$. Regarding the PMMA coverage, its

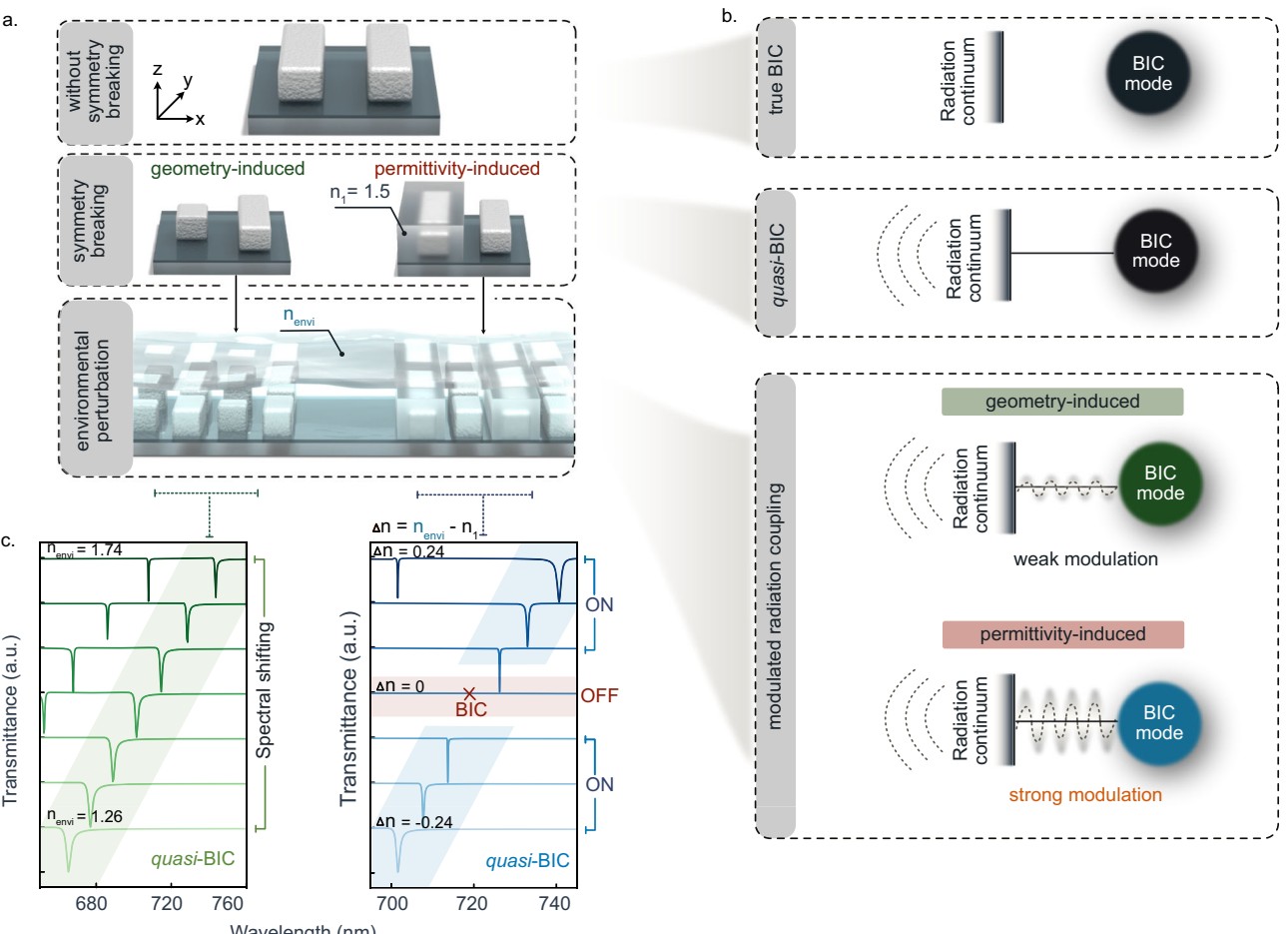

**Fig. 1 | Strong modulation of radiation coupling in permittivity asymmetric quasi-BIC metasurfaces under environment perturbation. a** Illustration of the transition from symmetry-protected BIC (top) to *q*BIC (middle) via two distinct approaches: geometry-induced and permittivity-induced symmetry breaking within the unit cell. Schematic of embedding these two kinds of asymmetric BIC metasurfaces into different environments (bottom), where the refractive index (RI) is variable. The changing of RI can be taken as the environmental perturbation for the permittivity asymmetric case, leading to different asymmetry parameters. **b** Schematic illustration of the BIC coupling mechanism. The symmetric geometry

with two identical dielectric rods corresponds to a true BIC state, exhibiting no coupling with the radiation continuum (top). Introducing a symmetry-breaking perturbation enables coupling to the far field and transforms the bound state into *q*BIC (middle), which present distinct optical responses to the environmental RI changes depending on the type of symmetry breaking (bottom). **c** Simulated transmittance spectra reveal that the geometry-induced *q*BICs only exhibit spectral shifts when environmental RI changes, while permittivity-induced *q*BICs respond dynamically and present a transition between the 'ON' and 'OFF' states.

predominant contribution arises from electric dipole moments $P_y$. Although its peak value is almost 5 times less than that of rods, it is enough to transform non-radiative BIC into a pronounced *q*BIC.

Geometry-driven *q*BICs with fixed asymmetry parameters after fabrication, as, e.g., determined by the geometrical differences between two resonators within the unit cell, can naturally also be modulated by surrounding RI changes, which leads only to shifts of the resonance position without direct control over the *Q* factor (Fig. 1c). Therefore, although useful in certain contexts, this mechanism does not fully exploit the potential for dynamic interaction with the surrounding environment. In contrast, our ε-*q*BIC metasurface design directly and strongly responds to perturbations in the surrounding RI, providing significant advantages over traditional g-*q*BICs. Simulation spectra provide clear evidence that, as the environmental RI transitions from 1.3 to 1.7, the ε-*q*BIC metasurfaces not only exhibit a pronounced shift in the optical response but also demonstrate a unique ability to turn the *q*BIC resonance 'ON' and 'OFF' in the transmission spectrum (Fig. 1c). This behavior is directly correlated with the RI contrast between the environments of the different rods, highlighting the superior adaptability of ε-*q*BICs over traditional g-*q*BICs.

## Experimental realization of reconfigurable environmental ε-*q*BIC

To construct ε-*q*BIC metasurfaces, we demonstrate a multi-step nanofabrication approach to obtain unit cells consisting of two identical rods embedded in distinct surrounding mediums. As shown in Fig. 2a, the procedure begins with the spin-coating of PMMA onto pre-fabricated symmetry-protected BIC metasurfaces (see "Methods" for fabrication details). The crucial stage of our fabrication strategy occurs from the second step, where the aim is to construct ε-*q*BIC meta-surfaces characterized by alternating rows of $TiO_2$ nano-rods. These rows are distinctively configured, with one set embedded in PMMA and the other exposed to air, thereby introducing a deliberate per-mittivity asymmetry essential for ε-*q*BIC functionality. To achieve that, a well-defined lithographic marker system is applied to provide precise spatial alignment.

Subsequent development of the exposed PMMA regions results in the formation of metasurfaces where each unit cell hosts two identical nano-rods, each within a distinct medium: either PMMA or air. The successful fabrication of PMMA-based ε-*q*BIC metasurfaces is confirmed by scanning electron microscopy (SEM) in Fig. 2b. The precise spatial alignment between the two lithographic steps is evidenced by a

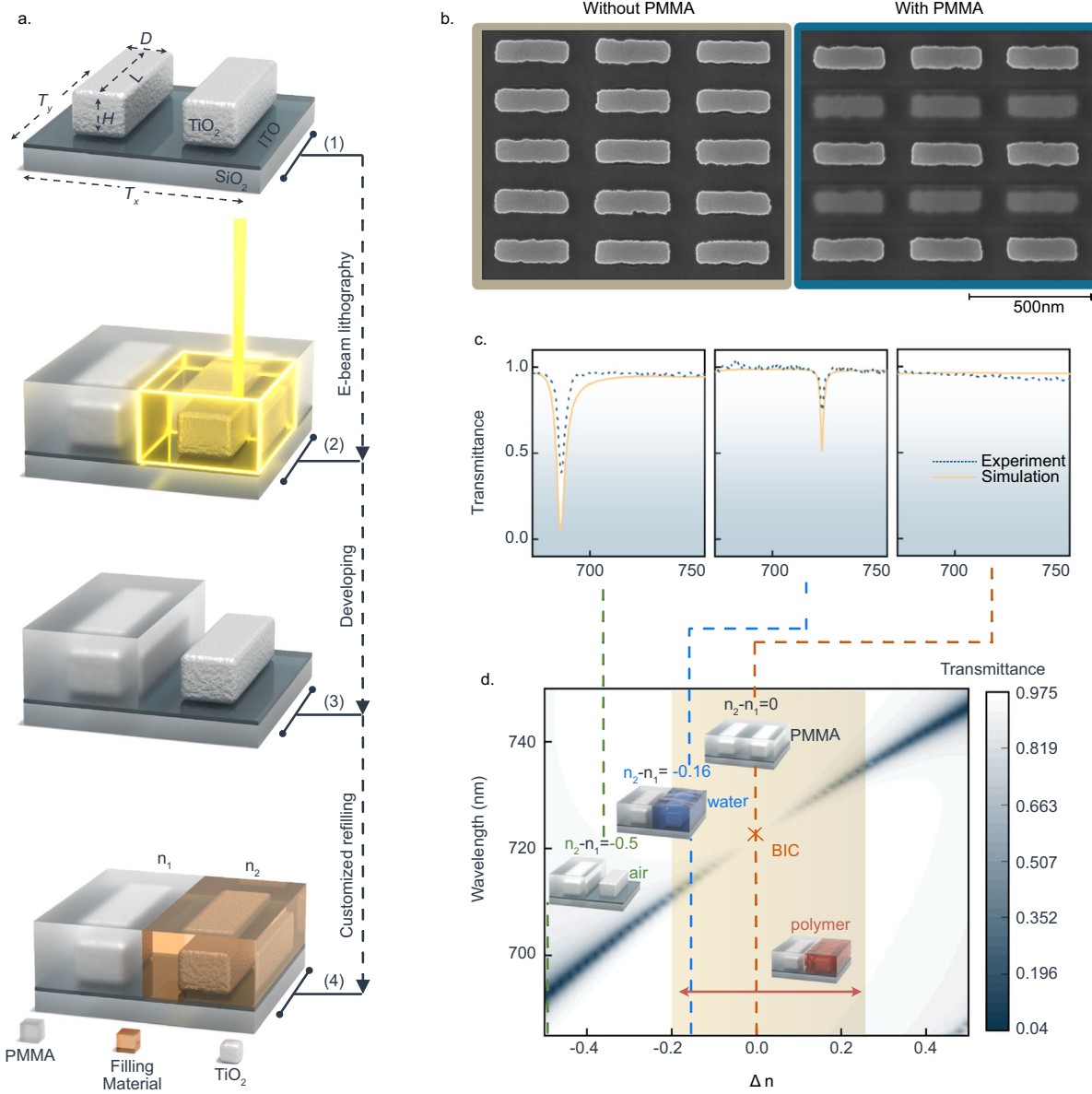

**Fig. 2 | Reconfigurable permittivity-asymmetry BIC metasurface demonstration. a** Workflow for the fabrication and post-fabrication tuning of ε-*q*BIC metasurfaces. The geometrical parameters of the unit cell are: $L = 321$ nm, $D = 107$ nm, $H = 140$ nm, $T_x = T_y = 403$ nm. The thickness of ITO is 50 nm. **b** SEM images of the fabricated metasurfaces corresponding to the true BIC without permittivity symmetry breaking (before PMMA coating), and the *q*BIC stemming from a permittivity asymmetry (after PMMA coating). **c** Experimental and numerical transmittance spectra confirm the high reconfigurability of ε-*q*BIC metasurfaces through customized refilling of different environmental media (air, water or PMMA) for tuning the refractive index contrast (△*n*) around the two rods. **d** Color-coded simulated transmittance map of different ε-*q*BIC metasurfaces as a function of Δ*n* and the wavelength. Inserts depict schematic illustrations of unit cells with two rods embedded into different materials. The orange shadowed region showcases the wide tunability of Δ*n* provided by the conductive polymer PANI as a surrounding medium (see Section 2.3).

distinct visual contrast in each unit cell. Specifically, the rod exposed to air displays markedly different features compared to its counterpart encased in PMMA. Here, the PMMA layer is applied with a thickness of 200 nm, sufficient to completely encase the nano-resonators, which are 140 nm in height. Supplementary Fig. 10 illustrates the influences of PMMA thickness on ε-*q*BICs in transmittance spectra, indicating that saturation gradually occurs after 140 nm. Beyond PMMA, this approach is applicable to a variety of resists that offer different RI contrasts (Supplementary Fig. 12), offering on-demand design flexibility.

The asymmetric factor of PMMA-based ε-*q*BIC metasurfaces is the RI contrast Δ*n*, which can be actively reconfigured by immersing it in different environments, such as air (Δ*n* = −0.5), water (Δ*n* = −0.17), and PMMA (Δ*n* = 0), resulting in different optical response as depicted in Fig. 2c. Specifically, pronounced ε-*q*BICs were observed in the transmittance spectra with the largest asymmetric factor (Δ*n* = −0.5). When the asymmetric factor is back to zero (PMMA environment), the quasi-BICs return to the true BIC state, as evidenced by the disappearance of the signal in the transmittance spectra.

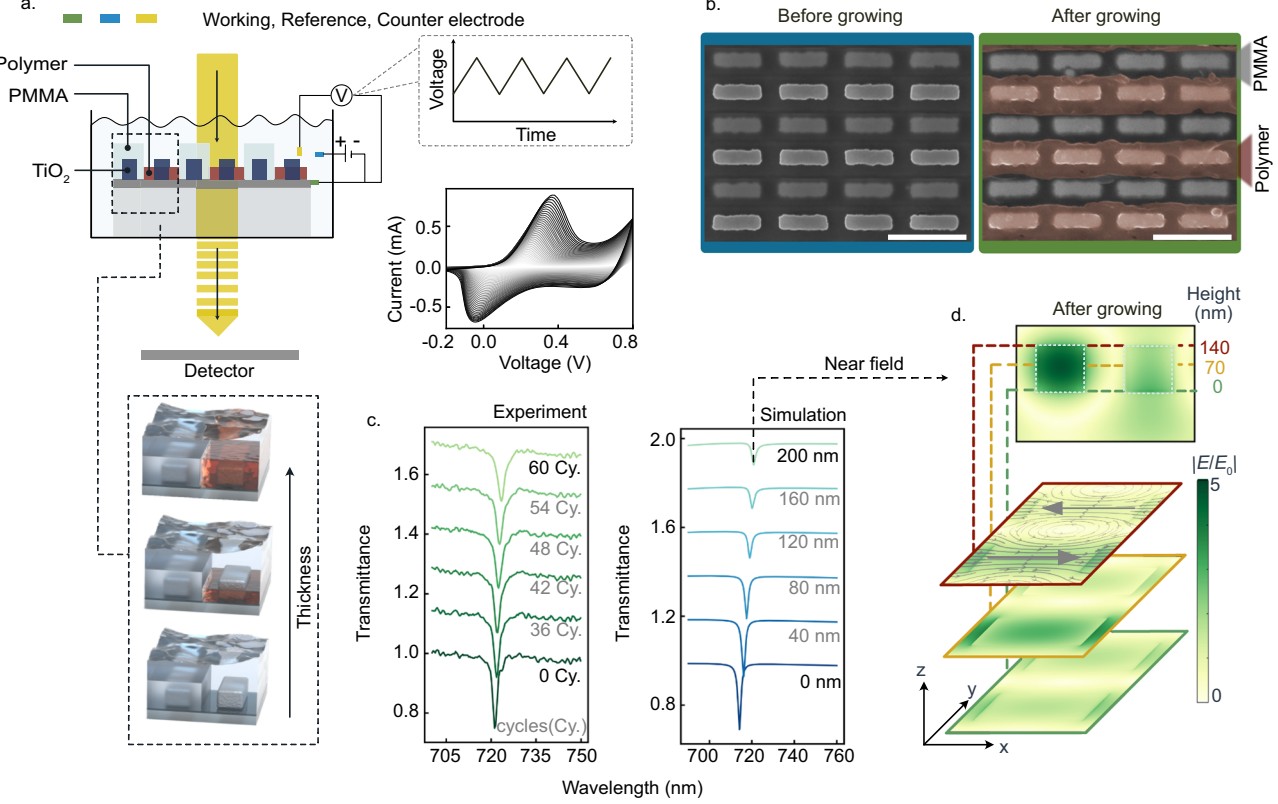

**Fig. 3 | In-situ coating of conductive polymer on the ε-qBIC metasurface.**
**a** Schematics of the in-situ polymer growth in an electrochemical cell. The polymer is grown over a conductive ITO layer without PMMA covering around the rod exposed to the aqueous environment (bottom inset). A scanning voltage from −0.2 to +0.8 V with a total of 60 cycles is applied to polymerize PANI and coat the nano-rods (right). **b** SEM images of the ε-qBIC metasurfaces before (left) and after (right) in-situ polymer coating (60 cycles). Scale bar: 500 nm. **c** Experimental in-situ transmittance measurement of ε-qBIC metasurfaces with different coating cycles

(0, 36, 42, 48, 54, 60) of the PANI at the reduced state (left). Simulated transmittance spectra of ε-qBIC metasurfaces with increasing thickness (0, 40, 80, 120, 160, 200 nm) of the PANI (right). **d** Simulated electric near fields of individual unit cells with 200-nm thick polymer covering on one of the nano-rods. Top: the side view (yz-plane) of the electric field distribution. Bottom: the sliced planer electric field distribution (xy-plane) at different heights (0 nm, 70 nm, 140 nm) corresponding to the two-rod unit cell.

In contrast to static alterations of the surrounding medium, the conductive polymer (PANI) enables dynamic tailoring of the optical response of this system (Fig. 2d). Its RI can be electrically tailored between 1.3 (oxidized state) and 1.7 (reduced state)[31].

**Conductive polymer in-situ coating on the ε-qBIC metasurface**
As a proof of concept, an electrochemical method was utilized to in-situ grow PANI on the bare resonators, while monitoring the optical transmission response in real-time (Fig. 3a). The PANI coating process was initiated through electrochemical polymerization utilizing an aqueous electrolyte. Here, the pre-deposited ITO layer underneath the metasurface functioned as the working electrode, complemented by an Ag/AgCl reference electrode and a Pt wire as the counter electrode. The polymerization was precisely controlled via a linear scanning voltage from −0.2 V to 0.8 V at a scanning rate of 25 mV/s. As PMMA covers every second row (odd rows) of the nano-rod resonators, blocking the electrolyte access, resonators in the complementary (even) rows remain exposed to the aniline electrolyte and accessible for row-selective PANI coating. After 60 cycles of coating, the PANI achieved a thickness comparable to that of the PMMA (200 nm). The successful integration of PANI onto the metasurface was confirmed through SEM in Fig. 3b, which indicates the preferential PANI coating on the even rows of resonators, while the PMMA continues to cover the odd rows.

This in-situ polymer coating approach allows for the real-time observation of the metasurfaces' optical response under continuous environmental perturbation throughout the PANI coating process. As shown in Fig. 3c, increasing PANI thickness on the PMMA-based ε-qBIC metasurfaces induces a red shift in the qBIC resonance alongside a reduction in modulation depth. Additionally, both the asymmetry factor $\Delta n$ change from −0.16 (between $n_{electrolyte} = 1.34$ and $n_{PMMA} = 1.5$) to 0.2 (between $n_{PANI} = 1.70$ and $n_{PMMA} = 1.5$) and the intrinsic loss of PANI reduce the $Q$ factor of the qBIC resonance. These trends agree well with the numerical simulations, indicating a consistent optical response to the polymer's accretion. Deviations from simulated transmittance spectra primarily stem from the inhomogeneity in the morphology of the coated PANI, which can act as a limit for such configuration. After completion of the PANI coating process, the system maintains its qBIC state, as evidenced by the $\Delta n$ (asymmetric factor) of 0.2 between PANI ($n = 1.7$ in the reduced state) with PMMA ($n = 1.5$). Simulated electric field distributions reveal the asymmetric mode pattern across the two TiO$_2$ nano-rods within each unit cell (Fig. 3d). By analyzing the $Q$ factors derived from the simulated transmittance spectra with increasing thickness of PANI, and correlating these with data from experimental in-situ PANI coating cycles, we explore the potential relationship between PANI thickness and its growth cycles (Supplementary Fig. 14).

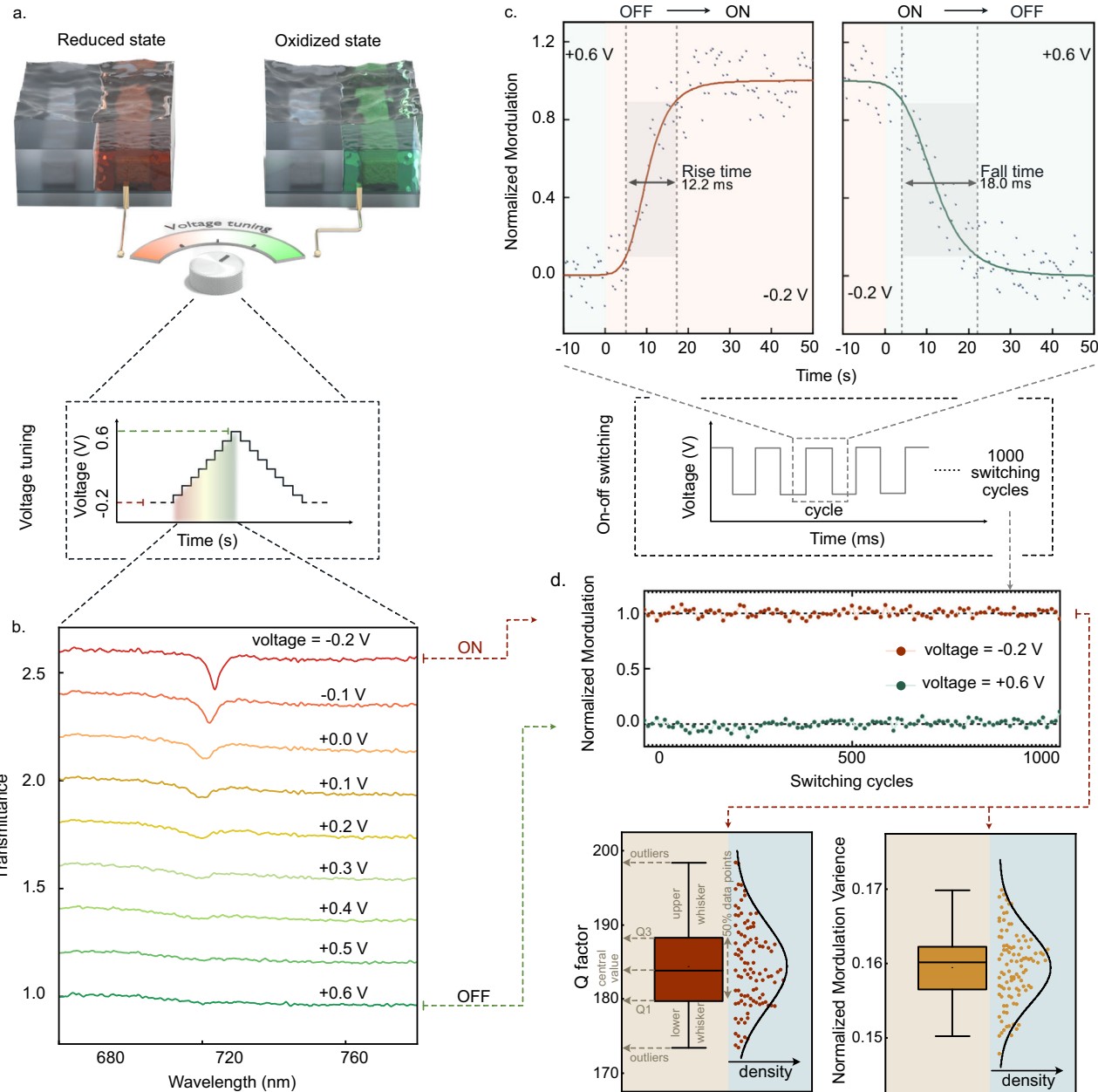

**Fig. 4 | Electrically tuning the optical properties of ε-qBIC metasurfaces.**
**a** Schematics illustrating the electrical tuning of the optical response of the ε-qBICs. The device is depicted in the reduced and oxidized states, corresponding to an applied voltage of −0.2 V and +0.6 V, respectively. The inset shows the voltage-time profile for tuning the intermediate states of the PANI. **b** Transmittance spectra at various applied voltages demonstrate the transition of ε-qBIC between high (ON) and low (OFF) transmittance states. **c** Time-resolved normalized modulation response of the electrical switching of the ε-qBIC. The rise time and fall time, defined by 10% to 90% modulation time for the ON/OFF switching process, are measured to be 12.2 ms and 18 ms, respectively. **d** Durability analysis of the electrical switching of the ε-qBIC. The upper graph presents the normalized modulation

of the ON/OFF states for over 1000 switching cycles. The lower graphs quantify the variations in the $Q$ factor and modulation of the ε-qBIC at the ON state for every 10 switching cycles in the 1000-cycle test. In the box plot, the box itself represents the interquartile range (IQR, IQR = Q3−Q1), which is the range between the first quartile (Q1, 25th percentile) and the third quartile (Q3, 75th percentile). Extending from Q1 to the smallest value within 1.5 IQR below Q1 is the lower whisker, and extending from Q3 to the largest value within 1.5 IQR above Q3 is the upper whisker. Data points outside the range of whiskers are considered outliers. Here, the curve inside is the kernel density estimation curve visually providing the comparison of the density on both sides of the median and also the density of the data points around that value.

After integrating PANI into PMMA-based ε-qBIC metasurfaces, the system's dynamic reconfigurability can be further explored by electrically adjusting PANI's refractive index.

## Electrically reconfigurable ε-qBICs based on polymer state switching

Leveraging the advanced electrochemical tunability of PANI, this conductive polymer stands out for its ability to undergo reversible

transitions between its reduced and oxidized states (Fig. 4a) with large-scale variation in RI (Supplementary Fig. 13)[31]. Different RI contrasts between PANI and PMMA ($n_{PMMA}$ = 1.5) can therefore control the radiation loss of this system. The extinction coefficient of PANI increases with the applied voltage[31], which will also interact with ε-qBIC resonance as the intrinsic loss channel.

The ε-qBIC resonance modulation strength can be effectively tailored by applying various voltages from −0.2 V to +0.6 V (Fig. 4b).

Specifically, the ε-$q$BIC resonance modulation strength decreases together with slight blueshifts in wavelength, which is in agreement with our design predictions. We define the ε-$q$BICs with the highest and lowest transmittance modulation as ON and OFF states, corresponding to the applied voltages of −0.2 V, and +0.6 V, respectively. We evaluate the switching speed of our ε-$q$BIC metasurface by measuring the time-resolved transmittance response while modulating the applied voltage between the ON (−0.2 V) and OFF (+0.6 V) states, as presented in Fig. 4c. The rise time and fall time are defined as the time required for the $q$BIC transmittance modulation to rise or fall between the 10% and 90% switching window for switch-on and switch-off processes, respectively, where the modulated intensity is defined as the normalized amplitude of the $q$BIC resonances. The rise time and fall time are measured to be 12.2 ms and 18 ms, respectively, which are mainly limited by the intrinsic material property of the PANI electro-optical dynamics[31]. To demonstrate the durability and repeatability of the electrical switching, we monitor the reconfigurable ε-$q$BICs in our system for 1000 switching cycles. It is clearly observed that the ε-$q$BIC at the ON and OFF states remain fully reversible even after 1000 switching cycles (Fig. 4d). The stable switching performance can be potentially further extended to $10^7$ switching cycles[44]. The key characteristic supporting this excellent durability is that, when electron injection or extraction takes place by voltage application, the highly conductive PANI can redistribute delocalized π-electrons along the polymer chain without structural degradation[45,46]. We further statistically confirm the consistency of the $Q$-factor and modulation of the ε-$q$BICs in the ON states of 100 cycles, evenly sampling from the total 1000 switching cycles. Both plots show that the bulk of the data is concentrated around the median, indicating the stability and consistency of the ε-$q$BIC metasurfaces during the electrical switching processes.

Our research introduces a new approach for engineering BIC metasurfaces via permittivity-induced symmetry breaking in the surrounding environment of the resonators, a significant departure from traditional, geometry-reliant methods. This strategy not only overcomes the inherent constraints of geometric asymmetry manipulation but also unlocks the flexibility of post-fabrication control over the coupling to the radiation continuum, a governing aspect of BIC physics. Additionally, our ε-$q$BIC metasurface design showcases enhanced optical response to the refractive index perturbations of the surrounding medium. In contrast to straightforward spectral shifts, our ε-$q$BIC metasurface concept opens up possibilities to finely modulate between *quasi* and true BIC states. We have experimentally demonstrated the generation of the ε-$q$BICs from specifically designed metasurfaces, where the coupling channel of BIC to continuum radiation is established by the embedding of two identical $TiO_2$ nano-rods into surrounding media with different permittivity (such as PMMA and air). The ε-$q$BICs concept is further exploited for realizing electrically reconfigurable BICs by integrating a conductive polymer into our metasurfaces. The advantageous electro-optical properties of PANI endow ε-$q$BICs with excellent electrical reconfigurability. By tuning the optical properties of PANI, the ε-$q$BICs can be dynamically switchable, exhibiting fully reconfigurable resonances with high modulation contrast. Additionally, we have demonstrated the electrical switching performance of the ε-$q$BICs, with fast switching speed (12.2 ms and 18 ms for switch-on and switch-off processes, respectively), and excellent switching durability even after 1000 cycles. Our design is compatible with a range of active materials beyond PANI, including liquid crystals, phase change materials, and transparent conducting oxides. The permittivity-induced BIC metasurfaces unlock an additional degree of freedom for manipulating and engineering BIC states, thereby laying the groundwork for future on-demand integrated electro-optical devices.

## Methods

### Numerical simulations

Numerical simulations of the permittivity-asymmetric $q$BIC metasurfaces were carried out with CST Microwave Studio with the frequency domain solver. The RIs of PANI in the reduced state and $TiO_2$ were taken from the data of in-house white-light ellipsometry (Supplementary Fig. 13). We utilized the default values implemented in CST Studio Suite for the $SiO_2$ and ITO. The refractive indexes of PMMA were taken from the standard database[47]. The simulations were performed within a rectangular spatial domain containing one unit cell with periodic boundary conditions. The multipole analysis was conducted in COMSOL Multiphysics by using the Electromagnetic Waves, Frequency Domain module, and Wavelength Domain study for one unit cell with the periodic boundary conditions on the sides.

### Nanofabrication

We apply sputter deposition (Angstrom) to deposit on the fused $SiO_2$ substrate with multiple layers of materials, which are in sequence 50-nm ITO and 140-nm $TiO_2$. The fabrication of the ε-$q$BIC metasurfaces was based on a three-step electron beam lithography (EBL) process. For the lithography steps, the sample was first spin-coated with a layer of photoresist (PMMA 950 K A4) followed by a conducting layer (ESpacer 300Z). We apply eLINE Plus from Raith Nanofabrication with an acceleration voltage of 20 kV and 15 μm aperture to the pattern on the photoresist layer.

In the first patterning process, a 30 nm thick gold marker system was defined on the top of $TiO_2$ film, which was used for aligning the following two fabrication steps. For the second patterning process, based on the gold marker system, we define the position to pattern the metasurfaces with two-rod nanostructures. Subsequent development is carried out in a 3:1 IPA: MIBK (methyl isobutyl ketone) solution for 135 s, followed by deposition of a 50 nm chromium layer as a hard mask. Lift-off was conducted in Microposit Remover 1165 overnight at 80 °C. Then, the sample with the hard mask was transferred for reactive ion etching (RIE) of the $TiO_2$ (140 nm) using a PlasmaPro 100 ICP-RIE from Oxford Instruments. After the RIE etching, the chromium hard mask was removed in a wet Cr etchant. In the last patterning process, we pattern the PMMA layer based on the same marker system on the fabricated BIC metasurfaces with the design of arrays of alternating grooves. Subsequently, we conducted the same developing recipe to clean the patterned region.

### Electrochemical cell

The electrochemical polymer coating and electrical switching were carried out in a custom-built electrochemical cell. The electrochemical cell is designed for housing a three-electrode system into a thin layer of aqueous electrolyte with an optical thickness of 1 mm on the top of the sample substrate, where a thin transparent glass was used to seal the electrolyte on the top, allowing for an optical transmission measurement through the metasurface. The sample substrate was used as the bottom sealing glass of the cell. The cell also features on the side the in-let and out-let for electrolytes to enable flow-in and flow-out for electrolyte replacement, which is driven by an electrical injector. The sample substrate was connected as the working electrode through a striped metal plate as the electrode contact, whereas a Pt wire and an Ag/AgCl electrode were used as the counter and reference electrodes, respectively. A potentiostat (CHI-760e) is used to apply voltage over time to perform electrochemical polymer coating and switching.

### In-situ PANI coating and electrical switching

In-situ PANI coating was carried out by an electrochemical polymerization method. The ITO substrate with the metasurface sample was connected as the working electrode. A cycling voltage in the range from −0.2 V to +0.8 V at a scanning speed of 25 mV/s was applied to the sample in an acidic aqueous electrolyte containing 1 M $H_2SO_4$ and

0.2 M aniline. The thickness of the coated PANI thickness can be controlled by the number of voltage scanning cycles[48].

For the electrical switching, the electrolyte in the electrochemical cell was replaced by an aniline-free aqueous electrolyte containing only 1 M $H_2SO_4$. To switch the PANI to the oxidized state and reduced state, constant voltages of +0.6 V and −0.2 V, respectively, were applied on the ITO substrate. For electrical switching and cycling, a cycling voltage in the range between −0.2 V and +0.6 V was used.

## Optical characterization

The refractive index of PANI was obtained from an ellipsometry measurement on an ellipsometer with dual-rotating compensators and a spectrometer (J.A. Woollam, M2000XI-210). A 100-nm thick PANI film electrochemically prepared on an ITO-coated glass substrate was measured with angle-variable spectroscopic ellipsometry at incident angles of 65º, 70º, and 75º, using a bare ITO-coated glass as blank reference.

The transmittance measurements of the metasurfaces were carried out with a Witec optical microscope comprising an air objective (20 X, NA = 0.4, Zeiss, Germany). Illumination was provided by a Thorlabs OSL2 white light source.

## Data availability

The data that support the findings of this study are available from Zenodo, https://doi.org/10.5281/zenodo.13285200.

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

## Acknowledgements
This work was funded by the Deutsche Forschungsgemeinschaft (DFG, German Research Foundation) under grant numbers EXC 2089/1–390776260 (to S.A.M.), and TI 1063/1 (to A.T.). We also acknowledge the Bavarian program Solar Energies Go Hybrid (SolTech) and the Center for NanoScience (CeNS). W.L. acknowledges the Alexander von Humboldt Foundation for the postdoctoral fellowship. S.A.M. additionally acknowledges the Australian Research Council and the Lee-Lucas Chair in Physics. Funded by the European Union (ERC, META-NEXT, 101078018 and EIC, OMICSENS, 101129734) (to A.T. and A.A.). The authors are grateful to Maxim Gorkunov for useful discussions. Views and opinions expressed are, however, those of the author(s) only and do not necessarily reflect those of the European Union, the European Research Council Executive Agency, or the European Innovation Council and SMEs Executive Agency (EISMEA). Neither the European Union nor the granting authority can be held responsible for them.

## Author contributions
Conceptualization, data collection, and writing: H.H. and W.L. Methodology, and data analysis: H.H., W.L., and A.A. Review and editing: R.B., S.A.M., A.T. Supervision: S.A.M. and A.T. All authors have approved the final manuscript and agree to be accountable for all aspects of the work, ensuring its accuracy and integrity.

## Funding

## Competing interests
The authors declare no competing interests.
