## [Peer Review File · Nature Communications]

Environmental permittivity-asymmetric BIC metasurfaces with electrical reconfigurabilityReviewers' comments:

Reviewer #1 (Remarks to the Author):

In the rapidly evolving field of nanophotonics, achieving precise spectral and temporal light manipulation at the nanoscale is a critical challenge. Traditional photonic bound states in the continuum (BICs) rely on geometric symmetry breaking, making them vulnerable to fabrication imperfections and limiting their resonance quality factor. We propose the concept of environmental symmetry breaking by embedding identical resonators into a medium with regions of contrasting refractive indexes, activating permittivity-driven quasi-BIC resonances (ϵ -qBICs) without altering the resonator geometry. Integration of the electro-optically active polymer polyaniline (PANI) enables electrically reconfigurable ϵ -qBICs, demonstrating rapid switching, durability, and enhanced optical response, thus expanding optical control applications.

-Could the authors provide a detailed explanation of the key fabrication processes?

-Can the author use electromagnetic field analysis to analyze the mechanism of qBICs resonance generation?

Reviewer #2 (Remarks to the Author):

Metasurfaces with quasi BICs have extremely high quality factor. In this work, the authors fabricated an electrically reconfigurable BIC metasurface by integrating electro-optically active polymer (PANI). The symmetry breaking depends on the refractive index, which can be tuned by an applied voltage. The electrical switching of quasi BIC resonance with switching speed of 12.2 ms/18 ms for switch-on/off processes was experimentally observed. It is not uncommon to transform BIC to quasi BIC by asymmetry refractive index distributions. Realizing reconfigurable BIC metasurfaces through electrically tunable refractive index polymer is interesting. I think this work can be published after addressing the following comments.

1. In Figure 3a, what is the rough surface on the upper surface of the unit cell of BIC metasurface? With thickness growth of PANI, the refractive index difference of surrounding mediums of two resonators decrease (changing from 1.5 between PMMA and air to -0.5 between PMMA to PANI). The quality factor of quasi BIC should increase with the refractive index difference decrease. In left panel of Figure 3c, it seems the quality factor decrease with thickness growth of PANI. Why the quality factor decrease?
2. In Figure 2, the quality factor and working wavelength of quasi BIC changing with different refractive index. However, in Figure 3 and 4, the resonance wavelengths of the fabricated metasurface remain basically unchanged with thickness growth or changing voltage? What is the wavelength and quality factor modulation range of the fabricated metasurface?
3. In the Nanofabrication part, the author only described the processing process in words. I suggest providing a detailed processing flow chart in the supplementary information.
4. In Figure S2, multipole analysis of a pair of nanospheres was provided to illustrate the influence of surrounding medium on the scattering characteristics the BIC metasurface. The spherical resonator is an ideal case. I suggest the author also provide multipole analysis based on the actual structure simulation.
5. To realized reconfigurable metasurface, what is the advantage of electro-optically active polymer (PANI) compared to other refractive index tunable material, such as liquid crystal, phase-change material?
6. Is 1000 cycles the limit of the designed metasurface? What happens to the switching performance of the metasurface after more than 1000 cycles? I think the 1000 cycles limit its application. Switching metasurface by electrically switchable polymer in previous work has proven to be stable for an extremely long time, with little to no degradation over >107 cycles (Science, 2021, 374(6567): 612-616.). I suggest the authors provide a detailed performance comparison (durability, switching speeds, modulation range, et al.) of their metasurface and other reconfigurable metasurfaces and illustrate the

advantages and application prospects of their metasurface.

Reviewer #3 (Remarks to the Author):

I have just read the paper by Hu et al. titled "Environmental permittivity-asymmetric BIC metasurfaces with electrical reconfigurability". The concept of BIC switching via refractive index variation is quite interesting and the paper well-done and carried out. I have some minor comments for improvements. However, I could not but notice that, beside the introduction of the combination with q-BIC, the concept described for metasurface modulation was already introduced, although for circular polarization steering, by Robin Kaissner et al. "Electrochemically controlled metasurfaces with high-contrast switching at visible frequencies." *Sci. Adv.* 7, eabd9450(2021). DOI:10.1126/sciadv.abd9450. Although the application is a bit different and the fall/rise times slightly worse, the geometry and realization of polymer stripes is staggeringly similar as well as the analysis of the device (but this last point is understandable). Yet, I was surprised not to see this work mentioned by the authors as a reference given that it is reported by a well-known group in the field. Considering that the novelty in the present paper is the addition of qBIC, I am not at all convinced that this is novel enough for *Nat. Communication*. I would be more inclined for a more specialized journal such as *ACS Photonics*. I am not implying that the concept is extremely interesting, I am simply pointing out that it seems rather incremental especially technologically speaking.

In any case, here are some more comments to improve the manuscript.

1. While the advantage of the environmental-induced qBIC in the modulation depth is quite convincing, it is not clear the advantage in terms of actual qBIC spectral width achievable. The authors mention in the introduction on the third page (line 11) that lithographic techniques compromise the resonator performance. While I do agree with them that fabrication uncertainties might hinder the performances, it is not clear to me how much this approach is improving and how much more it is reliable compared to simple asymmetry induced by fabrication. Indeed, the uncertainty in the refractive index of the polymer PANI and PMMA might affect the Q factor as well. But it is not known how much. I imagine that there are also oscillations in the polymer chain alignment that might affect. Can the authors comment? Could they compare the width obtained with lithographical asymmetry-induced q-BIC with those obtained by environmental change?
2. Regarding the switching times, the authors claim fast reconfigurability. 12 ms to a first look do not seem considerably faster per se, at least if not compared with other realizations (for example in the previously mentioned paper the authors achieve 35 ms, only a factor 3 worse). Is it fast given the small voltages applied or in absolute circumstances? Could the author compare their performance with the state of the art considering also the applied voltages?
3. In addition, the authors in the introduction to underline the speed of their device refer to the fastest switching time (the rise time), while in the conclusions it seems that the limiting dynamic is the fall time (18.8 ms). I would be conservative in the introductory statement.
4. In Figure 2c it seems quite surprising that the experimental qBIC seems to show a higher Q factor with respect to the simulated one. Can the authors comment and give numbers?
5. Again, when depositing the PANI polymer the authors report a redshift along with a reduced modulation depth of the BIC. Yet, while simulations show indeed a decrease of depth, the experiments show a broadening (not visible in the simulations) rather than a significant reduction of the depth. The authors attribute this deviation to the inhomogeneity in polymer morphology. This brings me back to my first comment. How do the authors support the statement that environmental BIC grant higher reproducibility than asymmetric fabricated designs, while the morphology is indeed pointed out to be the limit for the agreement here? The authors should clarify.

Reply to reviewers' comments

and a summary of the changes made in the revised manuscript

Reviewer #1

General statement:

In the rapidly evolving field of nanophotonics, achieving precise spectral and temporal light manipulation at the nanoscale is a critical challenge. Traditional photonic bound states in the continuum (BICs) rely on geometric symmetry breaking, making them vulnerable to fabrication imperfections and limiting their resonance quality factor. We propose the concept of environmental symmetry breaking by embedding identical resonators into a medium with regions of contrasting refractive indexes, activating permittivity-driven quasi-BIC resonances (ϵ -qBICs) without altering the resonator geometry. Integration of the electro-optically active polymer polyaniline (PANI) enables electrically reconfigurable ϵ -qBICs, demonstrating rapid switching, durability, and enhanced optical response, thus expanding optical control applications.

We thank the reviewer for fully supporting the publication of our revised work.

Reviewer comment:

1) Could the authors provide a detailed explanation of the key fabrication processes?

Our response:

We appreciate the reviewer's interest in the details of our fabrication process. Below, we describe the steps involved, as illustrated in the accompanying figure. The fabrication process is divided into three distinct phases:

(a) Initial Marker Fabrication:

The first step involves fabricating the gold (Au) marker system. These markers serve as crucial alignment references for subsequent patterning of the two-rod nanostructure within each unit cell.

(b) TiO₂ nano-rods fabrication:

Following the initial setup, we proceed to deposit and pattern the TiO₂ layer. Utilizing the previously established Au markers, we can achieve precise patterning within the designated regions through a three-point alignment operation using the Elineplus system.

(c) Selective Exposure Process:

In the final fabrication stage, we perform the last exposure, where the patterning region is again defined by the Au marker system. This ensures precise exposure of only half of each unit cell.

The critical steps of this process have been clearly outlined in the provided sketches.

Figure S2. Schematic overview of nanofabrication for PMMA-based environmental permittivity-asymmetric BIC metasurfaces.

Action taken:

We have added the schematic overview of nanofabrication for PMMA-based environmental permittivity-asymmetric BIC metasurfaces into the supporting information (SI) as **Figure S2**.

Reviewer comment:

2) Can the author use electromagnetic field analysis to analyze the mechanism of qBICs resonance generation?

Our response:

We thank the reviewer for the insightful suggestion to apply electromagnetic (EM) field analysis to reveal the mechanism of q BICs resonance generation. Based on the simulated electric field for the symmetric (without PMMA), and asymmetric (with PMMA) unit cells (**Figure S1**), it is evident that the PMMA coverage disrupts the symmetric electric field distribution across the two rods. This symmetry breaking is clearly depicted in the simulations.

Our detailed multipole analysis (refer to **SI Note 1**) indicates that the primary contribution to the q BICs resonance is from the rods through electric dipole resonance P_y , which obviously have the same shapes but opposite signs within each rod. Meanwhile, the role of the magnetic dipole M_x and electric quadrupole Q_{yz} moments cannot be neglected: their corresponding absolute value at quasi-BIC resonance is only 3 times less than the value of P_y . Regarding the PMMA coverage, its predominant contribution arises from electric dipole moments P_y . Although its peak value is almost 5 times less than that of rods, it is enough to transform non-radiative BIC into a well-pronounced *quasi*-BIC.

Figure S1. **Multipole analysis of asymmetric permittivity in q BIC metasurfaces.** **a.** Simulated electric field distribution in arb.u. over the middle cut of the nanorods at the perfect BIC (without PMMA coverage) and **b.** q BIC (with PMMA coverage) resonant wavelengths. **c.** Transmittance spectra of q BIC metasurfaces, and contributions of electric dipole P_y , magnetic dipole M_x and electric quadrupole Q_{yz} moments in arb.u. provided by the rod immersed in the PMMA (red lines), uncovered rod (blue lines), and PMMA itself (gray lines).

Action taken:

We have added the multipole analysis of our environmental permittivity-asymmetric BIC based on the simulation of electromagnetic field as **Figure S1**, and the related discussion has been added into **SI note 1**.

Reviewer #2

General statement:

Metasurfaces with quasi BICs have extremely high quality factor. In this work, the authors fabricated an electrically reconfigurable BIC metasurface by integrating electro-optically active polymer (PANI). The symmetry breaking depends on the refractive index, which can be tuned by an applied voltage. The electrical switching of quasi BIC resonance with switching speed of 12.2 ms/18 ms for switch-on/off processes was experimentally observed. It is not uncommon to transform BIC to quasi BIC by asymmetry refractive index distributions. Realizing reconfigurable BIC metasurfaces through electrically tunable refractive index polymer is interesting. I think this work can be published after addressing the following comments.

We thank the reviewer for supporting the publication of our revised work.

Reviewer comment:

1) In Figure 3a, what is the rough surface on the upper surface of the unit cell of BIC metasurface? With thickness growth of PANI, the refractive index difference of surrounding mediums of two resonators decrease (changing from 1.5 between PMMA and air to -0.5 between PMMA to PANI). The quality factor of quasi BIC should increase with the refractive index difference decrease. In left panel of Figure 3c, it seems the quality factor decrease with thickness growth of PANI. Why the quality factor decrease?

Our response:

We thank the reviewer for your detailed examination of Figure 3. In Figure 3a, the depicted rough surface represents the aqueous environment within an electrochemical cell, essential for the in-situ growth of polyaniline (PANI). This illustration is intended to highlight that PANI is synthesized directly in the electrolyte.

Regarding the refractive index differences, they shift from -0.16 (between the electrolyte with $n_{\text{electrolyte}} = 1.34$ and PMMA with $n_{\text{PMMA}} = 1.5$) to 0.2 (between PANI with $n_{\text{PANI}} = 1.70$ and PMMA with $n_{\text{PMMA}} = 1.5$) during the growing of PANI. This change enhances the asymmetry of the environment surrounding the resonators. Hence, with the increasing asymmetry factor, the quality factor should decrease as shown in Figure 3c. Additionally, the intrinsic loss of PANI will also reduce the quality factor of the system.

Action taken:

We added the following sentence in the revised manuscript to the section of 2.3, page 10 lines 5 to 8. “Additionally, the asymmetry factor, Δn changes from -0.16 (between $n_{\text{electrolyte}} = 1.34$ and $n_{\text{PMMA}} = 1.5$) to 0.2 (between $n_{\text{PANI}} = 1.70$ and $n_{\text{PMMA}} = 1.5$) accompanied with the intrinsic loss of PANI reduces the Q factor of quasi-BIC resonance.”

Reviewer comment:

2) In Figure 2, the quality factor and working wavelength of quasi BIC changing with different refractive index. However, in Figure 3 and 4, the resonance wavelengths of the fabricated metasurface remain basically unchanged with thickness growth or changing voltage? What is the wavelength and quality factor modulation range of the fabricated metasurface?

Our response:

Thank you for your valuable feedback on resonance shift and quality factor change of fabricated metasurfaces during the PANI growing process and electrical switching. During the growth of PANI (Figure 3), we observed a resonance shift of approximately 3 nm, which aligns closely with the simulation predictions of a 5 nm shift. This slight discrepancy can be attributed to the morphological inhomogeneity of PANI as it forms, which affects the uniformity and consequently the optical properties of the metasurface. Details of this are more clearly illustrated in **Figure S4a**. Regarding the quality factor (Q factor), it changes from 456 (before of PANI coating) to 252 (end of PANI coating), as detailed in **Figure S14**.

In the electrical switching process of PANI (Figure 4b), we observe a more significant resonance wavelength shift of about 10 nm, as shown in **Figure S4b**. This larger shift is a result of the substantial refractive index changes induced by the electrical modulation of PANI. The significant decrease of the Q factor can also be observed in **Figure S4b**.

Figure S5. The shifting of q BIC resonances during the PANI growing and switching processes. a. Experimental in-situ transmittance measurements of ϵ - q BIC metasurfaces with different PANI coating cycles (0, 36, 42, 48, 54, 60) in the reduced state. The transmittance shift of approximately 3 nm is indicated (color map corresponds to Figure 3c). **b.** Experimental transmittance spectra at various applied

voltages, demonstrating the transition of ϵ -qBICs between high (ON) and low (OFF) transmittance states. The resonance shift range is highlighted (color map corresponds to Figure 4b).

Action taken:

We have added the Figure S4 into the supporting information.

Reviewer comment:

3) In the Nanofabrication part, the author only described the processing process in words. I suggest providing a detailed processing flow chart in the supplementary information.

Our response:

We appreciate the reviewer's interest in the details of our fabrication process. Below, we describe the steps involved, as illustrated in the accompanying figure. The fabrication process is divided into three distinct phases:

(a) Initial Marker Fabrication:

The first step involves fabricating the gold (Au) marker system. These markers serve as crucial alignment references for subsequent patterning of the two-rod nanostructure within each unit cell.

(b) TiO₂ nano-rods fabrication:

Following the initial setup, we proceed to deposit and pattern the TiO₂ layer. Utilizing the previously established Au markers, we can achieve precise patterning within the designated regions through a three-point alignment operation using the Elineplus system.

(c) Selective Exposure Process:

In the final fabrication stage, we perform the last exposure, where the patterning region is again defined by the Au marker system. This ensures precise exposure of only half of each unit cell.

The critical steps of this process have been clearly outlined in the provided sketches.

Figure S2. Schematic overview of nanofabrication for PMMA based environmental permittivity-asymmetric BIC metasurfaces.

Action taken:

We have added the schematic overview of nanofabrication for PMMA based environmental permittivity-asymmetric BIC metasurfaces into the supporting information (SI) as **Figure S2**.

Reviewer comment:

4) In Figure S2, multipole analysis of a pair of nanospheres was provided to illustrate the influence of surrounding medium on the scattering characteristics the BIC metasurface. The spherical resonator is an ideal case. I suggest the author also provide multipole analysis based on the actual structure simulation.

We thank the reviewer for the insightful suggestion of applying multipole analysis to reveal the mechanism of qBICs resonance generation. Based on the simulated electric field for the symmetric (without PMMA), and asymmetric (with PMMA) unit cells (**Figure S1**) it can be clearly seen that the coverage of PMMA on one rod can break the symmetric electric field distribution in two rods. Through the detailed multipole analysis (**SI note 1**), the main contribution is given by the rods due to the electric dipole resonance P_y , which obviously has the same shapes and opposite signs within each rod before symmetry breaking. While, the role of the magnetic dipole M_x and electric quadrupole Q_{yz} moments cannot be neglected: their corresponding absolute value at quasi-BIC resonance is only 3 times less than the value of P_y . As for PMMA, its main contribution is due to the electric dipole moments P_y . Although its peak value is almost 5 times less than that of rods, it is enough to transform non-radiative BIC into a well-pronounced quasi-BIC.

Figure S1. **Multipole analysis of asymmetric permittivity in qBIC metasurfaces.** **a.** Simulated electric field distribution in arb.u. over the middle cut of the nanorods at the perfect BIC (without PMMA coverage) and **b.** qBIC (with PMMA coverage) resonant wavelengths. **c.** Transmittance spectra of qBIC metasurfaces,

and contributions of electric dipole P_y , magnetic dipole M_x and electric quadrupole Q_{yz} moments in arb.u. provided by the rod immersed in the PMMA (red lines), uncovered rod (blue lines), and PMMA itself (gray lines).

Action taken:

We have added the multipole analysis of our environmental permittivity-asymmetric BIC based on the simulation of the electromagnetic field into the supporting information (SI) as **Figure S1**, and the related discussion has been added to **SI note 1**.

Reviewer comment:

5) To realized reconfigurable metasurface, what is the advantage of electro-optically active polymer (PANI) compared to other refractive index tunable material, such as liquid crystal, phase-change material?

Our response:

We thank the reviewer for the insightful query regarding the choice of PANI over other tunable refractive index materials like liquid crystals and phase-change materials. Compared to other refractive index tunable materials, PANI has many desired properties such as large variation of refractive index, fast switching speed, superior cycling stability, and low operation voltages [*Adv Mater* 2017, 29 (8), 28004862]. Specifically, the electrical tunability with low operation voltages simplifies the device architecture and integration with other electronic components. And the excellent durability and stability of PANI contrasts with phase-change materials, which can suffer from changes in optical properties due to repeated phase transitions, and liquid crystals may be prone to alignment issues over time [*Adv Funct Mater*, 2019, 29 (10), 1806692]. However due to the rapid development in the field of reconfigurable metasurfaces, a wide variety of active materials have been applied for diverse functionalities, tailored to the specific application.

Reviewer comment:

6) Is 1000 cycles the limit of the designed metasurface? What happens to the switching performance of the metasurface after more than 1000 cycles? I think the 1000 cycles limit its application. Switching metasurface by electrically switchable polymer in previous work has proven to be stable for an extremely long time, with little to no degradation over >10⁷ cycles

(*Science*, 2021, 374(6567): 612-616.). I suggest the authors provide a detailed performance comparison (durability, switching speeds, modulation range, et al.) of their metasurface and other reconfigurable metasurfaces and illustrate the advantages and application prospects of their metasurface.

Our response:

We thank the reviewer for raising this question. We believe that 1000 cycles are not the limit of our designed environmental permittivity-asymmetric q BIC (ϵ - q BIC) metasurfaces. The switching durability

primarily depends on the dynamic properties of PANI. In a previously reported paper [Nanophotonics 2024, 13(1), 39], the PANI-integrated metasurfaces were shown to stably switch for over 2000 cycles without noticeable degradation. Another study [Adv. Mater. 2021, 33(41), 2103217] demonstrated stable switching performance can after 10^7 cycles. In our work, for the sake of time, we have shown that there is no significant degradation in the first 1000 cycles, which aligns with these previous findings. Therefore, we believe that our designed ϵ - q BIC metasurfaces possess superior switching durability, ensuring that switching stability will not significantly impact potential applications.

Detailed performance comparison between other reconfigurable metasurfaces is shown as below:

Chalcogenide PCMs: Reconfigurable metasurfaces using chalcogenide phase-change materials (PCMs) [Nat. Nanotechnol. 2021, 16(6), 661] exhibit fast switching speeds (5 μ s and 500 ms for switch-on and -off processes) but suffer from low modulation contrast (\sim 80%) and a switching duration of approximately 5×10^5 cycles.

III-V Semiconducting Materials: Metasurfaces using III-V semiconducting materials [Adv. Opt. Mater. 2014, 2(11), 1057-1063] feature very fast switching speeds (10 ns) and long, stable switching durations, but they have a disadvantage of low modulation contrast (\sim 30%).

Our PANI-Integrated ϵ - q BIC Metasurfaces: In our work, PANI-integrated ϵ - q BIC metasurfaces demonstrate relatively fast switching speeds (12 ms for switch-on and 18 ms for switch-off), superior switching stability (potentially up to 10^7 cycles), and large modulation contrast (almost 100%).

It is important to choose active materials for reconfigurable metasurfaces wisely, based on specific application requirements. From an application perspective, our electrically reconfigurable ϵ - q BIC metasurfaces offer advantages in dynamic switching q BICs with excellent overall performance in terms of switching speed, stability, and modulation contrast. These features make them highly suitable for integrated electro-optical devices.

Action taken:

We have added a sentence in the first paragraph after Fig. 4:

“...even after 1000 switching cycles (**Fig. 4d**). **The stable switching performance can be potentially further extended to 10^7 switching cycles.** The key characteristic supporting excellent durability is that...”

Reviewer #3

General statement:

I have just read the paper by Hu et al. titled “Environmental permittivity-asymmetric BIC metasurfaces with electrical reconfigurability”. The concept of BIC switching via refractive index variation is quite interesting and the paper well-done and carried out. I have some minor comments for improvements. However, I could not but notice that, beside the introduction of the combination with q -BIC, the concept described for metasurface modulation was already introduced, although for circular polarization steering, by Robin Kaissner et al. “Electrochemically controlled metasurfaces with high-contrast switching at visible frequencies.” Sci. Adv.7, eabd9450(2021). DOI:10.1126/sciadv.abd9450. Although the application is a bit different and the

fall/rise times slightly worse, the geometry and realization of polymer stripes is staggeringly similar as well as the analysis of the device (but this last point is understandable). Yet, I was surprised not to see this work mentioned by the authors as a reference given that it is reported by a well-known group in the field. Considering that the novelty in the present paper is the addition of qBIC, I am not at all convinced that this is novel enough for Nat. Communication. I would be more inclined for a more specialized journal such as ACS Photonics. I am not implying that the concept is extremely interesting, I am simply pointing out that it seems rather incremental especially technologically speaking. In any case, here are some more comments to improve the manuscript.

Our response:

We thank the reviewer for thoughtful comments and for bringing the work of “Electrochemically controlled metasurfaces with high-contrast switching at visible frequencies” to our attention, we have now added it as **ref.30**. We appreciate your acknowledgment of the interesting aspects of our research and commented as “The concept of BIC switching via refractive index variation is quite interesting and the paper well-done and carried out”.

While the approach of using electrochemically controlled polymers (PANI) for tuning optical properties, as detailed by Kaissner et al., bears similarities to our method in terms of the use of polymers and the general concept of refractive index modulation, there are several distinctive aspects of our work that underscore its novelty and impact.

(1) Kaissner et al. innovatively utilize PANI for phase profile control in plasmonic metasurfaces designed based on the Pancharatnam-Berry (PB) phase, and mainly focus on demonstrating the superior performance of this methodology, without bringing innovations to the fundamental physics of the PB phase or of plasmonic metasurfaces.

(2) However, we are working with dielectric metasurfaces rather than plasmonic ones. And PANI applied in our BIC metasurfaces does not merely modulate the optical phase or intensity but fundamentally alters the light-matter interaction mechanism at the nanoscale, providing fundamental insights on emergent resonances due to permittivity symmetry breaking.

(3) More importantly, our work starts by introducing a fundamentally novel concept by incorporating environmental permittivity asymmetry to active and control optical BICs, which present many advantages over the traditional symmetry-protected BIC metasurfaces that primarily rely on geometric manipulations to alter metasurfaces properties. Besides, we have applied a multipole analysis (**supplementary note 1**) to reveal the physical mechanism of qBIC generation, enriching the knowledge of the physics behind the symmetry-protected BIC. Furthermore, we experimentally demonstrate the feasible realization of this conceptual design with the integration of PMMA, a widely used material in the nanofabrication technique. Additionally, as a proof of concept, we integrate PANI into our platform as an electrically reconfigurable metasurface, presenting that our design is well adaptable to the popular active control technique. Hence, we are not presuming to demonstrate the superior performance of a PANI-based methodology which has been proved by Kaissner et al. in a PB phase control application. Alternatively, we emphasize more on a new insight into BIC physics with our design which can be validated both theoretically and experimentally.

From advancements in symmetry-protected BIC perspective:

(1) Our work introduces a fundamentally novel approach by incorporating environmental permittivity asymmetry to activate and control qBICs. Unlike traditional methods, which primarily rely on static

geometric manipulations to alter metasurface properties, our strategy employs a dual-material embedding technique (**Figure 1a**). This technique uses the intrinsic optical properties of different surrounding media to induce and manipulate qBIC resonances. This is a significant conceptual progress as it unlocks a new degree of freedom to manipulate BICs based on environmental conditions rather than just structural design.

(2) The use of environmental permittivity asymmetry enables an unprecedented level of control over the BIC metasurfaces' optical responses. By strategically selecting and altering the surrounding media, we can dynamically modulate the permittivity landscape across the metasurfaces. This enhances the tunability of the optical properties such as wavelength, modulation, and quality factor. More importantly, we can freely switch between qBIC and true BIC states (**Figure 1c**), which can't be achieved with traditional geometry-asymmetric BIC metasurfaces. This approach, in principle, allows significant improvement in the optical responses (modulation, Q-factors) when fabrication/material losses are further mitigated.

(3) The significant optical response to the environmental permittivity asymmetry of this design makes it a good candidate to be incorporated with electro-active materials as a reconfigurable platform extends its functionality beyond static optical elements. And its excellent adaptability to the modern active tuning technique for example PANI-based has also been presented (**Figure 3 and 4**).

To conclude, the conceptual innovations presented in our work not only advance the fundamental understanding of light-matter interactions in the BIC system but also pave the way for developing new optical devices.

Reviewer comment:

1) While the advantage of the environmental-induced qBIC in the modulation depth is quite convincing, it is not clear the advantage in terms of actual qBIC spectral width achievable. The authors mention in the introduction on the third page (line 11) that lithographic techniques compromise the resonator performance. While I do agree with them that fabrication uncertainties might hinder the performances, it is not clear to me how much this approach is improving and how much more it is reliable compared to simple asymmetry induced by fabrication. Indeed, the uncertainty in the refractive index of the polymer PANI and PMMA might affect the Q factor as well. But it is not known how much. I imagine that there are also oscillations in the polymer chain alignment that might affect. Can the authors comment? Could they compare the width obtained with lithographical asymmetry-induced q-BIC with those obtained by environmental change?

Our response:

We thank the reviewer for insightful comments regarding the spectral width and reliability of qBICs achieved through our environmental permittivity asymmetry approach. And we appreciate your acknowledgment to the advantage of our environmental permittivity-asymmetric BIC (ϵ -qBICs) metasurfaces.

Rather than directly compete with geometrical asymmetry metasurfaces in pushing quality factors from the technical fabrication perspective, our primary contribution lies in introducing a novel design concept that leverages environmental permittivity asymmetry to manipulate qBICs. This approach represents a paradigm shift from the conventional focus on geometrical modification. Here, we would like to put focus in broadening the conceptual understanding of how qBICs can be achieved and manipulated. We have conducted the multipole analysis to reveal the origination of qBICs in our design, and demonstrated the dominant contribution of electric dipole in this system (**supplementary note1**). Additionally, we also

present that environmental conditions play a unique and crucial role in enabling the transition between q BIC and true BIC states, which can't be realized in geometrical asymmetry BIC designs (**Figure 1**). The active responses to environmental permittivity variations of our design enable a higher degree of freedom in manipulating q BICs, which is a significant advantage over conventional geometrical asymmetry designs.

We agree with your concern about the uncertainty in the refractive index of the polymer PANI and PMMA, as well as oscillations in polymer chain alignment, which might affect the Q factor. However, the integration of PANI into our system is not intended to achieve an ultra-high Q q BIC resonance over the traditional geometry asymmetrical q BICs. Instead, as a proof of concept, providing the tunable environmental permittivity configuration allows for dynamic tuning of the optical characteristics of q BICs through changes in the surrounding medium's permittivity, even post-fabrication.

We also agree that a comprehensive comparison from a nanofabrication perspective is valuable, which is technically feasible, but it requires a careful definition of the asymmetry factors in both systems to ensure a fair comparison for Q factor calculation. To achieve that, we need to conduct a deeper analysis of our system to identify such asymmetry factors, which so far can't be straightforwardly expressed.

To clarify our focus, we have revised the introduction to emphasize that our research is centered on the extra functionality enabled by the environmental asymmetry design, rather than on providing a direct comparison with traditional methods.

Action taken:

We have adjusted the statement in the introduction part as follows:

“The generation of q BICs in geometrically modulated systems strongly relies on the geometrical design and precision of current lithographic techniques.^{19,20} Moreover, the fixed geometry of the resonators upon fabrication limits the possibilities of harnessing dynamic q BICs for potential applications in optical modulation^{21,22}, dynamic sensor²³, and light guiding.^{24,25}

An alternative solution involves leveraging the isotropic permittivity of unit cell constituents to induce q BICs resonances, a strategy that circumvents the need for precise modifications of geometric asymmetry.²⁶⁻²⁹ Here, the asymmetry parameter is governed by the difference in the intrinsic permittivity within the unit cell, allowing for the induction of q BIC resonances through perturbations in the permittivity symmetry of resonators.”

Reviewer comment:

2) Regarding the switching times, the authors claim fast reconfigurability 12 ms to a first look do not seem considerably faster per se, at least if not compared with other realizations (for example in the previously mentioned paper the authors achieve 35 ms, only a factor 3 worse). Is it fast given the small voltages applied or in absolute circumstances? Could the author compare their performance with the state of the art considering also the applied voltages?

Our response:

We thank the reviewer for insightful comments regarding the switching time. As the reviewer has noticed, our ϵ - q BIC metasurfaces achieve switching times of 12.2 ms (rise time) and 18.0 ms (fall time) with low operation voltages ranging from -0.2 V to +0.6 V, which are approximately three times faster than the 35 ms in the reported work (ref 30).

The applied voltages and the switching speed of reconfigurable metasurfaces are primarily dependent on the active materials used. As suggested, we have selected several representative electrically switchable materials commonly used in reconfigurable metasurfaces. These include chalcogenide phase-change materials (PCMs) [Nat. Nanotechnol. 2021, 16(6), 661], ionic conductive materials [Nano Lett. 2019(11), 19, 7988], electrochromic polymers [Science 2021, 374(6567), 612], and liquid crystals [Nat. Commun. 2020, 11(1), 3574], and compared their switching time and required applied voltages as below:

Active materials	Switch-on voltage	Switch-on time	Switch-off voltage	Switch-off time
Chalcogenide PCMs	<12 V	500 ms	20 V – 23 V	5 μ s
Ionic conductive materials	+1 V	~25 s	-2 V	~25 s
Electrochromic polymers (PEDOT)	+1 V	20.8 ms	-1 V	9.1 ms
Liquid Crystals	30 V	65 ms	0	40 ms
PANI (our work)	-0.2 V	12.2 ms	+0.8 V	18 ms

In our work, we used PANI as the active material, which is a type of electrochromic polymer. Compared to chalcogenide PCMs, PANI exhibits a shorter switch-on time but a longer switch-off time. Additionally, PANI requires smaller applied voltages and has a faster switching time compared to liquid crystals, as reported in previous studies. It also has smaller applied voltages and switch times compared to liquid crystals reported in a previous paper. However, it is worth pointing out that, in addition to switching time and applied voltage, the optical modulation contrast as well as switching stability are also important performances for reconfigurable metasurfaces. From that perspective, even though chalcogenide PCMs exhibit fast switching speeds but suffer from low modulation contrast (~80%) [Nat. Nanotechnol. 2021, 16(6), 661]. While, In our work, PANI-integrated ϵ -qBIC metasurfaces demonstrate relatively fast switching speeds, and large modulation contrast (almost 100%).

Reviewer comment:

3) In addition, the authors in the introduction to underline the speed of their device refer to the fastest switching time (the rise time), while in the conclusions it seems that the limiting dynamic is the fall time (18.8 ms). I would be conservative in the introductory statement.

Our response:

We thank the reviewer for pointing that out. We agree that the limiting dynamic is the fall time (18.8 ms). We have revised the sentence in the introduction to reflect this as “Specifically, we successfully engineer the radiative coupling of the ϵ -qBICs by leveraging the electro-optical response of PANI, where the qBICs resonance in the transmittance spectra can be switched between the “ON” and “OFF” states with a fast switching speed of **18.8 ms** within a low operation voltage range from -0.2 V to +0.6 V.”

Action taken:

We have revised the sentence in the introduction as “Specifically, we successfully engineer the radiative coupling of the ϵ -qBICs by leveraging the electro-optical response of PANI, where the qBICs resonance in the transmittance spectra can be switched between the “ON” and “OFF” states with a fast switching speed of **18.8 ms** within low operation voltage range from -0.2 V to +0.6 V.”

Reviewer comment:

4) In Figure 2c it seems quite surprising that the experimental qBIC seems to show a higher Q factor with respect to the simulated one. Can the authors comment and give numbers?

Our response:

We thank the reviewer for pointing this out. We apply the temporal coupling model theory to fit the spectrum and determine the Q factors of both experimental and simulated qBICs. When exposed to air (Figure S4a left), the Q factors for fabricated and simulated qBIC are 201 to 184, respectively, and 463 to 473 for exposure to water (Figure S4a right). The slightly higher Q factor of fabricated metasurfaces is due to the PMMA coverage ratio being slightly smaller in practice compared to the simulations (Figure S4c).

Figure S4. Quality factor analysis of simulated and experimental PMMA based ϵ -qBICs metasurfaces. a. Experimental and numerical transmittance spectra confirm the high reconfigurability of ϵ -qBICs metasurfaces through customized refilling of different environmental media (air, water) for tuning the refractive index contrast (Δn) surrounding media of the two rods (appending to Figure 2). **b.** Quality factors are extracted based on the fitting with temporal mode coupling theory. **c.** The variance of quality

factor in experimental fabrication compared to simulations is due to the slightly smaller coverage ratio of PMMA. The width of PMMA coverage on the unit cell was designed as half of the unit cell, which could be slightly different during the experiment because of the fabrication tolerances.

Action taken:

We have added this quality factor analysis into supporting information as Figure S4.

Reviewer comment:

5) Again, when depositing the PANI polymer the authors report a redshift along with a reduced modulation depth of the BIC. Yet, while simulations show indeed a decrease of depth, the experiments show a broadening (not visible in the simulations) rather than a significant reduction of the depth. The authors attribute this deviation to the inhomogeneity in polymer morphology. This brings me back to my first comment. How do the authors support the statement that environmental BIC grants higher reproducibility than asymmetric fabricated designs, while the morphology is indeed pointed out to be the limit for the agreement here? The authors should clarify.

Our response:

We thank review for the insightful comments on the PANI growing process. We would like to clarify that our paper does not claim that environmental BICs inherently grant higher reproducibility than asymmetrically fabricated designs. Instead, our focus is on presenting the novel concept of environmental permittivity-asymmetric BIC (ϵ -qBIC) metasurfaces, which provide a new method for manipulating BICs through environmental changes rather than static geometric modifications. As mentioned in the introduction part, we use PANI here as an experimental demonstration to showcase the adaptability of our design to popular active tuning techniques.

Indeed, our experimental in-situ transmittance spectrum of PANI growth showed a small reduction in qBIC modulation, which was less significant than in the simulations. We attribute this discrepancy to the intrinsic inhomogeneity of the polymer growing on the surface, whereas in the simulations, we assume an ideal, continuous, and flat increase of PANI.

This reminds us to keep exploring better active materials to be integrated into our platform, which could also inspire more interesting findings in this field. We appreciate the opportunity to clarify this point and thank the reviewer for their valuable feedback, which has helped us improve the clarity of our manuscript.

Action taken:

We have added a sentence in the main text to state the limitation of PANI integration in our system, as “Deviations from simulated transmittance spectra primarily stem from the inhomogeneity in the morphology of the coated PANI, which can act as a limit for such configuration.”

REVIEWERS' COMMENTS

Reviewer #1 (Remarks to the Author):

The author has resolved all the issues raised and recommends acceptance.

Reviewer #2 (Remarks to the Author):

The authors have answered my questions properly.

Reviewer #3 (Remarks to the Author):

I have now read the response to the referee reports and the new version of the manuscript. I understand the point of the authors (which is stressed in various points) that is that they "introduce a novel design concept that leverages environmental permittivity asymmetry to manipulate qBICs" and I agree that it can have technological impact on optical modulation of metasurfaces possessing BIC modes. Yet, I again stress that the conceptual use of PANI to reconfigure metasurfaces is already there and the combination with BIC modes, although very sound, seems like an incremental step. The paper is well written and experiential results sound and well-supported. Therefore this paper deserves publication in a Photonic Journal. Yet, I still stand with my early impression that I am not convinced that it deserves publication in Nature Communication.

Reply to reviewers' comments and a summary of the changes made in the revised manuscript

Reviewer #1

General statement:

“The author has resolved all the issues raised and recommends acceptance.”

Our response:

We thank the reviewer for fully supporting the publication of our revised work.

Reviewer #2

General statement:

“The authors have answered my questions properly.”

Our response:

We thank the reviewer for supporting the publication of our revised work.

Reviewer #3

General statement:

“ I have now read the response to the referee reports and the new version of the manuscript. I understand the point of the authors (which is stressed in various points) that is that they "introduce a novel design concept that leverages environmental permittivity asymmetry to manipulate qBICs" and I agree that it can have technological impact on optical modulation of metasurfaces possessing BIC modes. Yet, I again stress that the conceptual use of PANI to reconfigure metasurfaces is already there and the combination with BIC modes, although very sound, seems like an incremental step. The paper is well written and experiential results sound and well-supported. Therefore this paper deserves publication in a Photonic Journal. Yet, I still stand with my early impression that I am not convinced that it deserves publication in Nature Communication.”

Our response:

We appreciate your recognition of the novelty of our work, and thank you for the comments “**The paper is well written and experiential results sound and well-supported.**” We believe that the innovative aspect of our work goes beyond the methodology design (integrating PANI). It opens new avenues for research and applications in photonics, which aligns well with high-impact and broad readership of Nature Communications.